# Towards Visual Text Design Transfer Across Languages

**Yejin Choi**[*]    **Jiwan Chung**[*]    **Sumin Shim**    **Giyeong Oh**    **Youngjae Yu**

Yonsei University

yejinchoi@yonsei.ac.kr

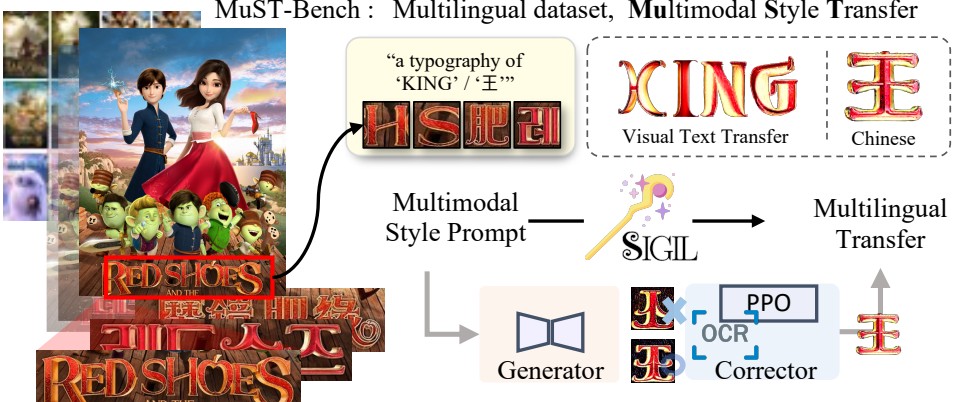

Figure 1: Generating multilingual visual text following a prompt with typography in the style of an input image from SIGIL for a multilingual film poster. The film poster is from the movie "Red Shoes and the 7 Dwarfs" by Locus Corporation.

## Abstract

Visual text design plays a critical role in conveying themes, emotions, and atmospheres in multimodal formats such as film posters and album covers. Translating these visual and textual elements across languages extends the concept of translation beyond mere text, requiring the adaptation of aesthetic and stylistic features. To address this, we introduce a novel task of Multimodal Style Translation (MuST-Bench), a benchmark designed to evaluate the ability of visual text generation models to perform translation across different writing systems while preserving design intent. Our initial experiments on MuST-Bench reveal that existing visual text generation models struggle with the proposed task due to the inadequacy of textual descriptions in conveying visual design. In response, we introduce SIGIL, a framework for multimodal style translation that eliminates the need for style descriptions. SIGIL enhances image generation models through three innovations: glyph latent for multilingual settings, pre-trained VAEs for stable style guidance, and an OCR model with reinforcement learning feedback for optimizing readable character generation. SIGIL outperforms existing baselines by achieving superior style consistency and legibility while maintaining visual fidelity, setting itself apart from traditional description-based approaches. We release MuST-Bench publicly for broader use and exploration[1].

---

[*]denotes equal contribution

[1]https://huggingface.co/datasets/yejinc/MuST-Bench

38th Conference on Neural Information Processing Systems (NeurIPS 2024) Track on Datasets and Benchmarks.

# 1 Introduction

Recent advancements in image generation have significantly enhanced the ability to create and manipulate text within images, enabling instruction-based control over text styles. However, achieving this level of fine-grained style manipulation becomes challenging in cross-language contexts. For instance, imagine the title text on a sci-fi movie poster, initially rendered in English with a futuristic, metallic font, being translated to Korean while preserving the same font style, color gradients for visual impact. Effective multimodal translation requires two factors: transferable visual content across languages and accurate multilingual text generation that maintains the original design.

To tackle this issue, we introduce MuST-Bench, a carefully curated dataset that compares language pairs with similar stylistic features across diverse typographical styles. This dataset enables robust evaluation of models' abilities to capture visual text design nuances across languages. Furthermore, each dataset entry is augmented with human-annotated bounding boxes at the character level, substantially enhancing the benchmark's quality and precision. MuST-Bench samples feature natural artistic typographies sourced from film posters, enabling the evaluation of transfer capabilities from English to five target languages with diverse writing systems: Chinese, Korean, Thai, Russian, and Arabic.

However, current text-to-image models generally rely on textual descriptions for image generation, which restricts their effectiveness in practical visual contexts such as film posters, album covers, or advertisements, where meticulously crafted artistic composition is crucial to communicate the main theme and atmosphere. In such cases, abstract concepts as *design* and *style* are hard-explained through text alone. As we empirically demonstrate later (see section 5.2), state-of-the-art visual text generation models struggle with our benchmark, failing to capture the intricate designs present in artistic typography instances.

To address the challenges, we introduce the Style Integrity and Glyph Incentive Learning (🪄SIGIL ) framework. SIGIL introduces two technical novelties: First, SIGIL guides the character generation process using a distance metric defined in the pre-trained glyph latent space. This space, based on a pre-trained VAE, allows for the comparison of glyphs across different languages. Our empirical results demonstrate that guidance within the glyph latent space achieves superior style fidelity compared to textual conditioning. Second, we employ an OCR model to generate confidence scores for the images during the generation process. These scores serve as rewards in a reinforcement-learning-based optimization, enhancing the readability of the generated text.

Specifically, our contributions are threefold:

1. **MuST-Bench**: We present MuST-Bench, a novel corpus designed for the task of multimodal style translation. MuST-Bench includes human-annotated character-level bounding boxes and encompasses a variety of languages (Chinese, Korean, Thai, Arabic, and Russian) along with diverse typographical styles, facilitating a thorough cross-linguistic evaluation of model performance.
2. 🪄SIGIL  framework: We propose Style Integrity and Glyph Incentive Learning (SIGIL) that enhances style transfer fidelity by using glyph latent guidance. By leveraging input style images for direct style guidance, SIGIL achieves superior consistency in generating styled text across different languages. It also incorporates a reinforcement learning approach that leverages rewards from an off-the-shelf OCR model, improving the letter accuracy of the generated images.
3. **Evaluation metrics**: We introduce a comprehensive evaluation scheme for multimodal style transfer tasks, assessing the robust transfer of both characters and styles. Our proposed metrics include model-based evaluation with an OCR model, image-to-image similarity scoring, and semantic evaluation using multimodal large language models. Notably, the semantic evaluation aligns closely with human evaluation results.

Table 1: Comparison of MuST-Bench with other datasets based on features.

| Dataset Name | Multi-Languages | Image paired | Style similarity | Character-level bbox |
|---|:---:|:---:|:---:|:---:|
| LAION-5B [20] | ✓ | ✗ | ✗ | ✗ |
| MARIO-10M [5] | ✗ | ✗ | ✗ | ✓ |
| AnyWord-3M [24] | ✓ | ✗ | ✗ | ✗ |
| MuST-Bench | ✓ | ✓ | ✓ | ✓ |

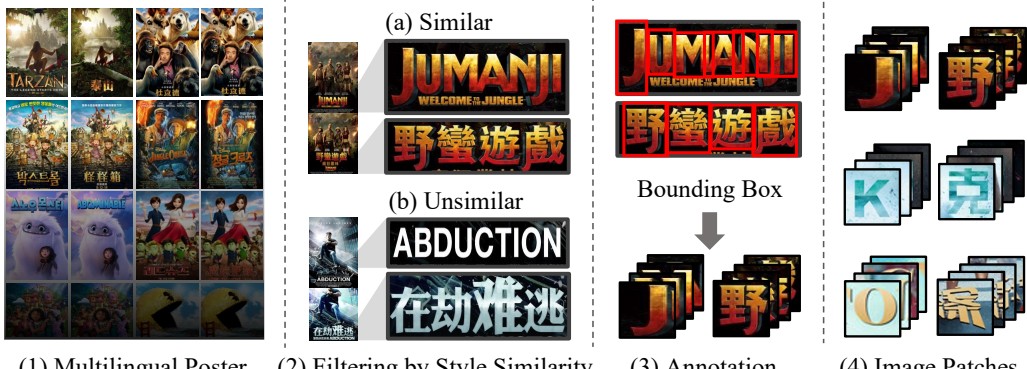

| (1) Multilingual Poster | (2) Filtering by Style Similarity | (3) Annotation | (4) Image Patches |

Figure 2: Overview of our data curation process: (1) For each film poster, we collect multilingual pairs. (2) We manually filter these pairs to retain those with similar typographic styles. (3) Character-level bounding boxes are manually annotated for each image. (4) Finally, we extract the character set pairs.

## 2 Related Work

**Text to Image Generation.** Recent advancements in text-to-image generation tasks [3, 7, 19], notably with models like DALLE-3 [3], have demonstrated exceptional capabilities in image synthesis. Despite these advancements, generating precise and coherent text within images remains challenging. To improve text coherence and accuracy, models like Imagen [19] and DeepFloyd IF [6] employ pre-trained large language models such as T5-XXL [16]. Stable Diffusion 3 [7] introduces a novel transformer-based architecture that uses separate weights for image and text tokens, enhancing text comprehension. However, these methods require an extensive image-caption pair dataset, making it difficult to extend them to languages other than English.

**Visual Text Generation.** Delving in text to image generation model's visual text generation ability, TextDiffuser [5] incorporates a Layout Transformer capable of understanding text positions and layouts, which helps the Diffusion Model generate text more effectively. GlyphControl [26] advances text generation in diffusion models by aligning text based on its location, implicitly considering font size and text box position. AnyText [24] uses text glyph, position, and masked image inputs alongside OCR-encoded stroke data and image caption embeddings, enabling multilingual visual text generation with a dataset of large-scale multilingual text images. However, further research is required to advance the concept of 'text as images' focusing on generating textual content that is itself viewed as a creative and visually appealing image. While several works [20, 5, 24] datasets provide valuable visual text data, they may not fully meet the requirements of our proposed task, as indicated in table 1. This discrepancy highlights the need for a new benchmark that is specifically tailored to address the unique demands of our research in visual text analysis.

**Subject-driven Image Generation.** In the domain of subject-driven image generation, models such as Textual Inversion[8], Dreambooth [18], and Custom Diffusion [11] have significantly advanced the ability to generate images consistent with specific artistic styles or visual prompts by utilizing images as inputs. However, they exhibit limitations in text generation. DS-Fusion [23] is a semantic typography generation method that integrates styles and glyphs through generative adversarial training with input style images. While it effectively achieves semantic typography generation, it can only train one glyph per subject. SIGIL expands this capability by allowing the training of two glyphs per subject and addresses issues in generating characters with diffusion models by employing OCR models in reinforcement learning for improvements.

**Cross-language text generation.** CLASTE [25] enabled cross-language scene text generation but requires a source-target language pair dataset. To create this dataset, synthetic data was generated by a rule-based tool for training. The training method for this data has not been disclosed, complicating further comparisons. Based on the published images, it appears the goal was to generate text similar to scene text rather than text with complex design.

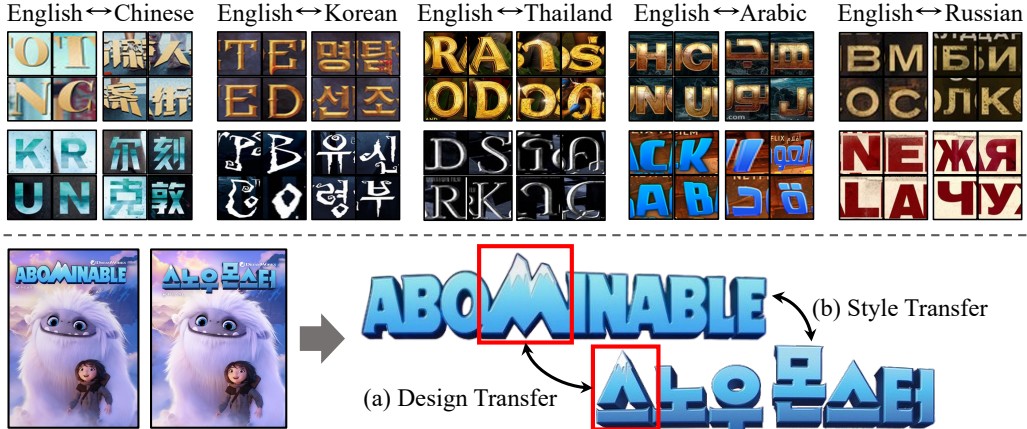

Figure 3: Examples of different language pairs in MuST-Bench. Style translation in MuST-Bench encompasses both (a) instance-level design transfer, illustrated by a snow-covered mountain, and (b) set-level style transfer, featuring a palette of blues, bold and angular fonts, and gradient textures. The film poster is from the movie "Abominable" by DreamWorks Animation, Pearl Studio.

## 3    MuST-Bench

**Corpus.**    To the best of our knowledge, MuST-Bench is the first dataset to feature instance-level multilingual typography pairs. Using English as the pivot language, we curated 432 film poster pairs for Korean, 427 pairs for Chinese, 100 pairs for Russian, 100 pairs for Thai, and 60 pairs for Arabic. Each film poster image is decomposed into $10 \sim 20$ typography character images. MuST-Bench encompasses a diverse range of typography styles across approximately 1,120 film poster images, as illustrated in fig. 3.

**Curation.**    Our data curation process, depicted in fig. 2, involves sourcing multilingual film poster images from a movie database website[2]. By using natural sources, we obtain instance-level typography pairs that capture the artistic choices of human designers. We then apply a rigorous human filtering process to select only cross-language posters that exhibit similar styles, pairing them accordingly. Subsequently, each character and whole word is manually labeled with precise bounding boxes. We plan to make our dataset publicly available. Refer to Appendix C for further details.

**Benchmark.**    We define multimodal style translation as a sequence-to-sequence character generation problem. Given an indexed list of characters from a film poster, the task is to generate a corresponding sequence of characters in the target language that closely matches the ground truth film poster in that language. Both the input and output sequences are formatted as cropped patches of characters within the image. Additionally, we allow the textual form of the characters as inputs, as illustrated in fig. 4.

**Evaluation metrics.**    Assessing the accuracy of style translation to the target language is a complex task. MuST-Bench simplifies this evaluation by providing ground-truth targets for each language. Given the target images, we compare their similarity to the generated character images. To evaluate the quality of style transfer, we utilize semantic image-to-image similarity with the off-the-shelf CLIP model [15], commonly referred to as CLIP-I in the literature [18]. Another critical criterion is the shape correctness of the generated visual text, i.e., readability. Following previous studies [24], we use an off-the-shelf OCR model to assess this accuracy metric. The OCR model transcribes the generated characters into text, which is then compared with the ground-truth text. The generated images are resized to 25 by 25 pixels as inputs for the OCR model, as larger characters are not interpretable by these models. For a fair comparison, we use a different OCR model for evaluation (PP-OCRv3 [13]) than the reward model used in training following previous literature [1].

While CLIP-I assesses general image-to-image similarity, we cannot control them to focus on typographic details. To address this, we propose using Multimodal Large Language Models (MLLMs)

---

[2]https://www.cinematerial.com/

for semantic comparisons of typography styles. These models are prompted to provide a Likert 5-point scale score [14]. Our initial experiments indicate that including both the ground truth and generated images enhances the accuracy of the model's evaluation. Therefore, we use both images as inputs in all experiments. For robustness, evaluations were conducted using both GPT-4V [28] and Claude-3-OPUS [2]. Further details are provided in Appendix A. Finally, to validate the evaluation metrics, we conduct human evaluations. Participants rate the style fidelity and shape correctness of the generated images using a Likert 5-point scale.

## 4 🪄SIGIL Framework

In the training process, we use 3 to 5 specific style images, which is a common number used in personalized image generation [18, 8]. Along with these style images, we provide a prompt, e.g., "a typography of 'K'" and a corresponding glyph image, e.g., a white background with a black 'K' glyph created using the Pillow library. However, the ground truth, e.g., the desired style image of the 'K' is not used during training. Since we assume there is no ground truth in real-world translation tasks, we introduce the SIGIL method, which allows us to generate the glyph without relying on direct supervision from ground truth.

At inference time, only the prompt, e.g., "a typography of 'K'," is used as input. Our model is trained on one style across multiple glyphs. As shown in Appendix G, while previous approaches [23] could only generate a single glyph per style, we have extended this capability to generate two glyphs. For instance, we can now generate both 'A' and 'B' in a vivid style. The training time for this process is detailed in Appendix D.

### 4.1 Overview

SIGIL utilizes three types of inputs to generate the sequence of character images in the target language. First, the sequence of character images in the source language is provided to guide typography styles. Second, the textual form of the target characters is rendered as glyphs to guide the target shape. Finally, the same textual forms are also provided as instructions using various prompt templates. These different prompts help control variations in the output characters. For further details, please refer to the experiments in Appendix B.

### 4.2 Glyph Guidance on the Latent Space

We extend a conditional diffusion model architecture [18] to transform character images from the source language (style prior) into styled character images in the target language. Following the parameterization by Ho et al. [9], we train a U-Net architecture $M_\theta$ to estimate the Gaussian noise $\epsilon_t$ at step $t$ using mean squared error (MSE) as the distance metric.

$$\mathcal{L}_{\text{diff}} = \|\hat{\epsilon} - \epsilon\|_2^2 = E_t[\|\hat{\epsilon}_t - \epsilon_t\|_2^2], \quad \hat{\epsilon}_t = M_\theta(z_{t+1}) \tag{1}$$

We adopt a latent diffusion setting [17], where the diffusion process occurs in the latent space $z$ defined by a pre-trained variational autoencoder (VAE) [10] encoder, rather than in the pixel space.

To incorporate glyph shape constraints, we propose an efficient solution: using the same VAE encoder to minimize the difference between the generated image and the rendered glyphs in the latent space. Specifically, the glyphs are encoded as latent vectors $z_g$ to be compared with the model-generated diffusion noise $\hat{\epsilon}_t$. Our *glyph* loss is defined as follows:

$$\mathcal{L}_{\text{glyph}} = \|\hat{\epsilon} - z_g\|_2^2 = E_t[\|\hat{\epsilon}_t - z_g\|_2^2] \tag{2}$$

By using the latent semantic space as our metric space, we provide a cost-effective way to compare characters across languages without training a separate classifier. As previous literature suggests [27, 12], representation spaces defined by pre-trained neural networks often better correlate with human perception. By using the same VAE for both diffusion framework and guidance loss, we provide an efficient solution for controlling glyph shapes. We demonstrate the effectiveness of latent space distances through qualitative ablation studies in fig. 7.

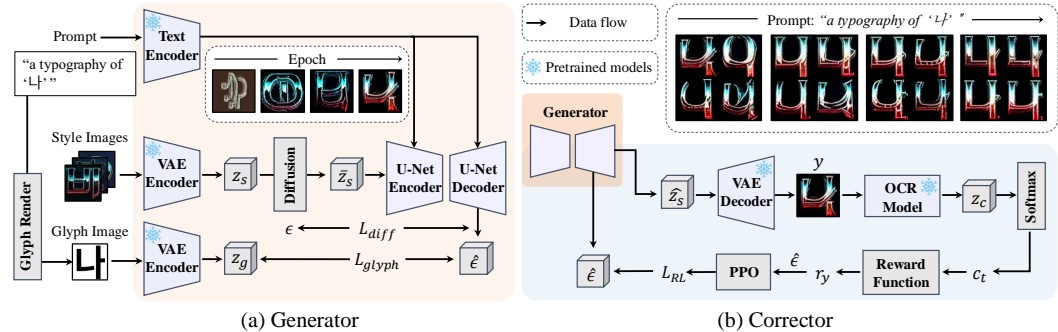

|     | (a) Generator |     | (b) Corrector |
| --- | --- | --- | --- |

Figure 4: SIGIL comprises two main components: the generator and the corrector. (a) the generator combines the style prior and the glyph guide on the VAE representation space to construct the target character and (b) the corrector exploits the off-the-shelf OCR model to optimize the readability of the generated character.

We combine both losses with a dynamic coefficient function $f(t)$ to arrive at the final objective $\mathcal{L}(t)$ per timestep $t$ of the diffusion process. The coefficient function $f(t)$ dynamically adjusts the influence of the glyph latent guidance based on the diffusion process stage.

$$\mathcal{L}(t) = \mathcal{L}_{\text{diff}} + f(t) \cdot \mathcal{L}_{\text{glyph}}, \quad f(t) = \lambda\left(1 - e^{-t/\tau}\right) \tag{3}$$

Through experimental optimization, we set the hyperparameters to $\lambda = 0.1$ and $\tau = 50$, balancing style preservation and fidelity. $\tau$ determines the rate of increase in the glyph loss influence during the training process. We further explore the effects of different $\lambda$ and $\tau$ values in section 5.3.

## 4.3 Enhancing Readability through OCR Rewards.

Whether two visual texts contain the same characters can be only poorly measured in the general image-based distance space. As a response, we propose to use off-the-shelf optical character recognition (OCR) models to provide goodness scores on how well the drawn character conforms to the original textual form. A hypothetical image $y_c$ corresponding to character $c$ is obtained by decoding the generated latent $\hat{z}_c$ of the U-Net module through the VAE decoder $f_v$. Given the image $y_c$, the OCR model $f_o$ outputs probabilities over the entire set of possible characters $c' \in C$. The probability of the label $c$ is used as a reward $R_{base}$ to optimize the U-Net parameters $M_\theta$.

$$R_{base}(\hat{z}_c) := f_o(c|y_c), \quad y_c = f_v(\hat{z}_c) \tag{4}$$

We identified a failure mode where the naive OCR rewards cause the generator to produce multiple small instances of the same letter to exploit the reward system. To counteract this issue, we propose a simple modification to the reward definition: introducing a penalty based on the number $n$ of detected characters. The character count $n$ is derived from the inference results of the same OCR model. The final reward function combines the base reward with a character number penalty $d(n)$:

$$R(\hat{z}_c) := R_{base}(\hat{z}_c) \cdot d(n), \quad d(n) = e^{-(n-1)} \tag{5}$$

We optimize the above criterion using a reinforcement learning approach. Specifically, we adopt Proximal Policy Optimization (PPO) [21] enhanced with engineering choices of Black et al.,[4]. Denoting the log probability of the next timestep in the backward diffusion process as $\log p(z_{t-1}|z_t)$, the final reinforcement learning loss $\mathcal{L}_{\text{RL}}$ is defined as:

$$\mathcal{L}_{\text{RL}} = \max\left(\mathcal{L}_t, \text{clip}(\mathcal{L}_t, 1 - \epsilon_{\text{clip}}, 1 + \epsilon_{\text{clip}})\right) \tag{6}$$

$$\mathcal{L}_t = -\text{clip}(A_t, -A_{\text{clip}}, A_{\text{clip}}) \cdot \exp(\log P_{\text{c},t} - \log P_{\text{i},t}) \tag{7}$$

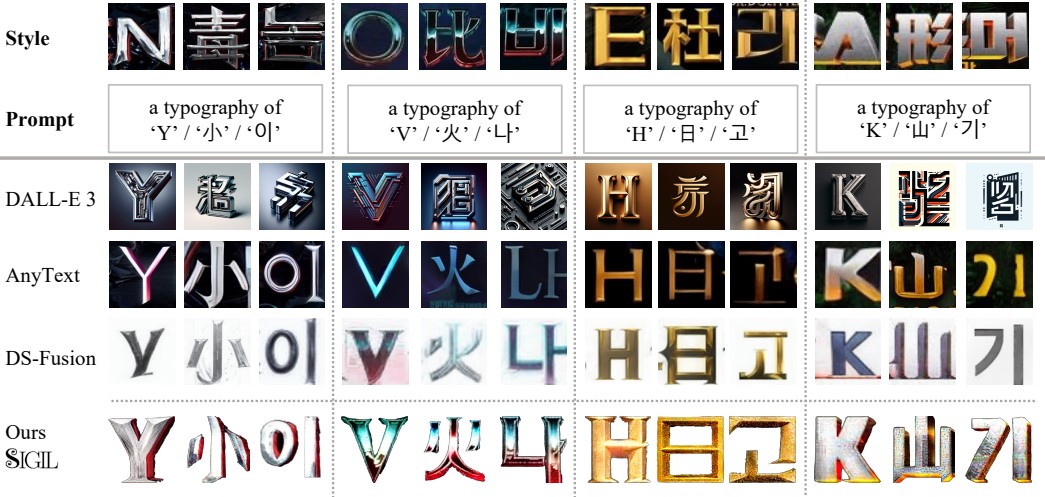

Figure 5: Visual comparison of state-of-the-art models and APIs in multilingual visual text generation. All input style images are selected from the MuST-Bench. All generated images were created using unseen text contents.

where $A_t$ is the advantage for timestep $t$, computed as:

$$A_t = \frac{R(\hat{z}_c) - \mu_R}{\sigma_R + \epsilon_{\text{adv}}} \tag{8}$$

where $\mu_R$ is the mean reward, $\sigma_R$ is the standard deviation of the rewards, and $\epsilon_{\text{adv}} = 1e - 8$. The advantage $A_t$ is then clipped between $-A_{\text{clip}}$ and $A_{\text{clip}}$, where $A_{\text{clip}} = 5$ and $\epsilon_{\text{clip}} = 1e - 4$. The log probabilities $\log P_{\text{c},t}$ are computed during the DDIM [22] step with the current model, while $\log P_{\text{i},t}$ are obtained from the initial log probability computation as described in Black et al., [4].

## 5 Experiment

### 5.1 Qualitative Evaluation

**Qualitative comparison**   We compared the latest models in the field of text-to-image generation, including DALL-E3, the multilingual text generation model AnyText, and the generation model DS-Fusion, focusing on their performance in generating English, Chinese, and Korean typography. To enable these models to accept input style images as references, we utilized GPT-4V as a bridge for DALL-E3, allowing it to receive style images (refer to Appendix E). AnyText facilitated this through its text editing capabilities. DALL-E3 produced visually impactful results, generating highly accurate text in English. However, it was limited to English and could only guide the image style through text prompts. Consequently, when comparing the input style image with the generated image, the visual similarity was not as strong. AnyText demonstrated high accuracy in generating text for the languages it was trained in, namely English, Chinese, and Korean. Despite this, the output's color, shape, and texture were relatively monotonous. DS-Fusion employed a method where style words and text to be generated were input as text prompts, creating images through Latent Diffusion Models (LDMs)[17] for model training. We compared the results from this method with those generated by directly inputting style images into the model for training. SIGIL excelled in creating textures and colors, faithfully replicating the style of the input images. For example, the first column of fig. 5, accurately reproduced the red border color and texture of the "Metal" style. In the second column, it captured the glossy texture and gradient colors of the "Cyber" style. In the third and fourth columns, it respectively reproduced the 3D shapes and textures of the "Gold" and "Vintage" styles, showing high fidelity to the input style images.

Table 2: Quantitative comparison using OCR and CLIP-I.

| Method | OCR ↑ | | | CLIP-I ↑ | | |
|---|---|---|---|---|---|---|
| | English | Chinese | Korean | English | Chinese | Korean |
| DALL-E3 | 0.4360 | 0 | 0 | 0.8210 | 0.8237 | 0.8255 |
| AnyText | 0.3333 | 0.3548 | 0.3245 | **0.8731** | **0.8688** | 0.8671 |
| DS-Fusion | 0.4808 | 0.2471 | 0.1906 | 0.8596 | 0.8531 | 0.8615 |
| SIGIL | **0.7163** | **0.7481** | **0.6577** | 0.8673 | 0.8618 | **0.8692** |

## 5.2 Quantitative Evaluation

**OCR and CLIP-I Evaluation.**   From the MuST-Bench, we derived 10 distinct style subjects and composed 26 prompts per language. Each prompt was sampled 4 times, generating a total of 1,040 styled characters per language. The evaluation utilized the benchmark and metrics discussed in section 3, and the results are shown in table 2. SIGIL significantly outperforms the others in OCR accuracy for English, Chinese, and Korean. For the CLIP-I score, AnyText achieved slightly higher scores for some languages. This is because AnyText uses image editing to generate results, which means that not only the text but also the background is included in the similarity calculation. Consequently, we remove the background for both human and MLLM-based evaluation.

Table 3: Multimodal Large Language Model Evaluation using Likert scale from 1 to 5.

| Evaluator | GPT-4V | | | Claude | | |
|---|---|---|---|---|---|---|
| | English | Chinese | Korean | English | Chinese | Korean |
| DALL-E3 | 3.00 | 2.12 | 2.11 | 2.70 | 2.30 | 2.30 |
| AnyText | 2.44 | 2.88 | 2.56 | 2.50 | 3.20 | 2.50 |
| DS-Fusion | 1.89 | 2.12 | 2.22 | 2.40 | 2.50 | 2.40 |
| SIGIL | **3.89** | **3.50** | **4.00** | **3.60** | **3.30** | **3.50** |

**MLLM Evaluation.**   For the style fidelity evaluation using MLLMs, we employed the method mentioned in section 3. To ensure the reproducibility of the evaluation results, we used a consistent input judgment template. For GPT-4V, we set the seed to 100 and the temperature to 0. Although Claude-3-OPUS does not support seed fixation, setting the temperature to 0 minimizes variability. To further ensure reproducibility, we provided consistent input data and verified the results across multiple runs.

For each evaluation, the images generated by each method were randomly shuffled before being presented to the MLLM for assessment. A total of 30 questions were involved in the evaluation process. The evaluation results are presented in table 3. SIGIL achieved the highest style fidelity scores on both GPT-4V and Claude. Interestingly, even though images were generated from DALL-E3 using prompts created by GPT-4V based on the input style image, SIGIL still received higher scores in the GPT-4V evaluation. During the evaluation, we identified a safety misclassification issue with GPT-4V for some typographic images, and four such samples are included in Appendix A.

Table 4: User study using ranking score.

| Method | Style fidelity | | | Legibility | | |
|---|---|---|---|---|---|---|
| | English | Chinese | Korean | English | Chinese | Korean |
| DALL-E3 | 0.1981 | 0.1760 | 0.1895 | 0.2620 | 0.1596 | 0.1624 |
| AnyText | 0.2156 | 0.2346 | 0.2238 | 0.2086 | 0.2661 | 0.2255 |
| DS-Fusion | 0.1809 | 0.1851 | 0.1797 | 0.2282 | 0.2174 | 0.2309 |
| SIGIL | **0.4054** | **0.4042** | **0.4069** | **0.3012** | **0.3568** | **0.3812** |

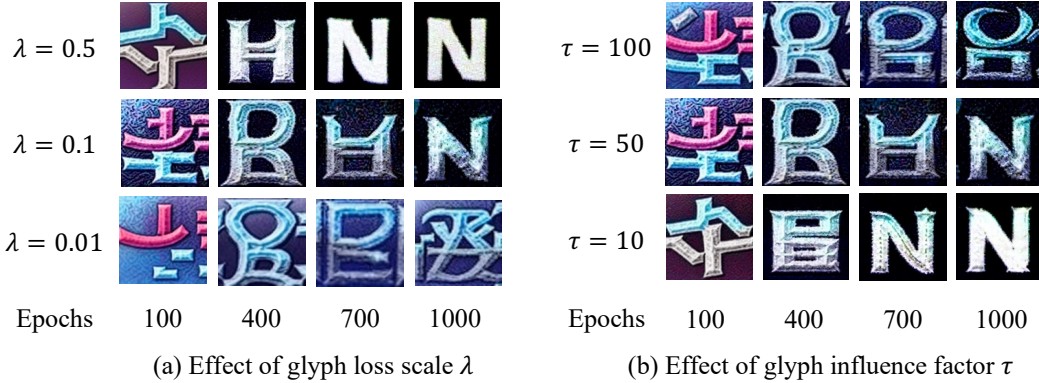

|  | Epochs 100 400 700 1000 |  | Epochs 100 400 700 1000 |
|---|---|---|---|
|  | (a) Effect of glyph loss scale $\lambda$ |  | (b) Effect of glyph influence factor $\tau$ |

Figure 6: Effect of glyph loss scale $\lambda$ and glyph influence factor $\tau$. The target character is *N*.

**User study.** We had human workers rank the model outputs based on style fidelity and legibility. A total of 60 users participated, using the same set of samples as in the MLLM evaluation. For the style fidelity evaluation, participants were instructed to *Rank the images in the order of similarity to the reference image*. For the legibility evaluation, they were asked to *Order the images by how well the visual text matches the prompt*. table 4 shows that SIGIL significantly outperforms the baselines. DALL-E3 scored second-highest in terms of legibility for English, but the lowest for all other languages. Despite its high resolution and visual effects, its low style fidelity score indicates difficulties in visual transfer based on text-only descriptions. For example, see the rows labeled "Style Images" and "DALL-E3" in fig. 5.

## 5.3 Ablation Study

We examine the impact of hyperparameters on the glyph loss coefficient $f(t)$. As shown in fig. 6 (a), higher values of $\lambda$ result in a rapid convergence of the glyph loss, but this often leads to a significant divergence from the intended style. Conversely, higher values of $\tau$ promote stable convergence, with the variations in generated results between epochs diminishing, as illustrated in fig. 6 (b).

Table 5: Quantitative results for glyph loss scale $\lambda$ and glyph influence factor $\tau$.

| $\lambda$ | OCR | CLIP-I |
|---|---|---|
| SIGIL($\lambda = 0.5$) | **0.9423** | 0.8565 |
| SIGIL($\lambda = 0.1$) | 0.7163 | 0.8673 |
| SIGIL($\lambda = 0.01$) | 0.0038 | **0.8855** |

| $\tau$ | OCR | CLIP-I |
|---|---|---|
| SIGIL($\tau = 100$) | 0.0663 | **0.8696** |
| SIGIL($\tau = 50$) | 0.7163 | 0.8673 |
| SIGIL($\tau = 10$) | **0.9276** | 0.8603 |

To further support our finding in fig. 6, we report quantitative results in table 5, which re-affirms our findings in the qualitative ablation studies.

In the generator component of SIGIL, if glyph images are used directly without glyph latent guidance, the generator training results are as shown in fig. 7 (a). In comparison, by using the glyph latent guidance we propose, the generation results can be obtained as shown in fig. 7 (b).

Table 6: Ablation study: quantitative results.

| Method | OCR ↑ | | | CLIP-I ↑ | | |
|---|---|---|---|---|---|---|
| | English | Chinese | Korean | English | Chinese | Korean |
| SIGIL(w/o $\mathcal{L}_{\text{glyph}}$) | 0.001 | 0.001 | 0.002 | **0.8864** | **0.8833** | 0.8834 |
| SIGIL(w/o Corrector) | 0.4529 | 0.3962 | 0.2188 | 0.8674 | 0.8617 | 0.8694 |
| SIGIL(w/o Generator) | 0.18 | 0 | 0 | 0.8078 | 0.801 | 0.804 |
| SIGIL | **0.7163** | **0.7481** | **0.6577** | 0.8673 | 0.8618 | **0.8692** |

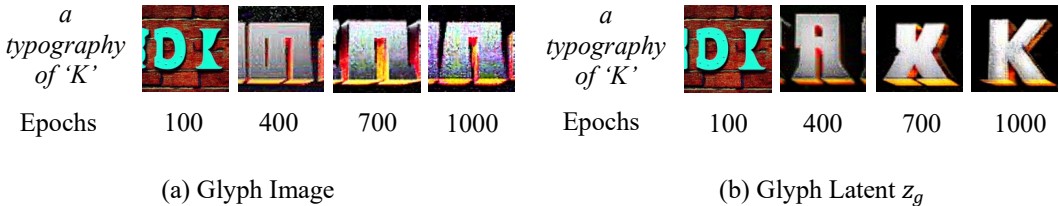

(a) Glyph Image (b) Glyph Latent $z_g$

Figure 7: Effectiveness of latent space distances: Generation results for the prompt "a typography of 'K'".

Table 7: Ablation study: multimodal large language model evaluation.

| Evaluator | GPT-4V | | | Claude | | |
|---|---|---|---|---|---|---|
| | English | Chinese | Korean | English | Chinese | Korean |
| SIGIL(w/o $\mathcal{L}_{\text{glyph}}$) | 2.5 | 2.6 | 2.5 | 2.9 | 3.2 | 2.6 |
| SIGIL(w/o Corrector) | 2.5 | 2.6 | 3.1 | 3.3 | 3.1 | 2.6 |
| SIGIL(w/o Generator) | 2 | 1.3 | 1.2 | 2.1 | 2.4 | 2.3 |
| SIGIL | **4.1** | **3.3** | **3.8** | **3.4** | **3.2** | **3.6** |

Additionally, we conduct more ablation studies on the objective part. Specifically, we test the effects of applying SIGIL without $\mathcal{L}_{\text{glyph}}$, SIGIL without the Generator (which combines style and glyph), and SIGIL without the Corrector (the RL part) to assess the contribution of each component.

The results in the table 6 show that the generator has the greatest impact on style generation accuracy, while glyph generation accuracy is most significantly influenced by $\mathcal{L}_{\text{glyph}}$. Additionally, the Corrector primarily proves effective when the generator already achieves a certain level of success in glyph generation. Also, the results presented in table 7 further demonstrate that the generator has the greatest influence on style fidelity.

# 6 Conclusions

In this paper, we introduce the SIGIL, a novel approach for generating typography in various languages while maintaining a consistent style. SIGIL can optimize generated images to match the input style image even with a small amount of data, without relying on extensive image-caption datasets. In this study, we have successfully combined two specific glyphs per style during training. In future work, we plan to explore methods that enable the combination of a greater number of glyphs.

Our contributions include a comprehensive benchmark to support the training and evaluation of multimodal style translation. The key advantages of our approach are as follows: (i) Unlike traditional visual text generation methods, our style subject-driven approach allows for effective model training without the need for large volumes of visual text data for each language. (ii) We propose an innovative technique that integrates an OCR model into the reinforcement learning process for generative models, significantly enhancing text generation accuracy. (iii) By providing a dataset and benchmark of multilingual text pairs in the same style, we establish a solid foundation for future research in expanding visual text generation across multiple languages. These contributions collectively advance the field of multilingual visual text generation, paving the way for further developments and applications in this area.

# 7 Acknowledgements

This work was partly supported by an IITP grant funded by the Korean Government (MSIT) (No. RS-2020-II201361, Artificial Intelligence Graduate School Program (Yonsei University) and RS-

2024- 00353131) and the National Research Foundation of Korea (NRF) grant funded by the Korea govern-ment (MSIT) (No. RS-2024-00354218).

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

# A MLMM Evaluation Details

## A.1 Comparison of Judgement Templates.

In our study, we explored two distinct methodologies for evaluating the output of MLLMs. The first approach consolidates all generated images into a single composite image, facilitating an evaluative process from a human-centric perspective. This approach is delineated in the judgment template shown in fig. 8. Alternatively, the second method involves separately encoding each generated image, which is illustrated in the judgment template of fig. 9.

Figure 8: Judgment template from a human perspective.

Figure 9: Judgment template for encoding each image.

The results of these evaluations are presented in fig. 10 and fig. 11, respectively. When employing the first methodology, wherein multiple images are assembled within a single frame, the visual comparisons and subsequent justifications of ratings demonstrate a high degree of consistency. Conversely, the approach of encoding and evaluating each image individually reveals variations in the visual assessments and the rationales provided for the ratings, indicating a divergence in consistency compared to the composite image evaluation method.

### Input Image:

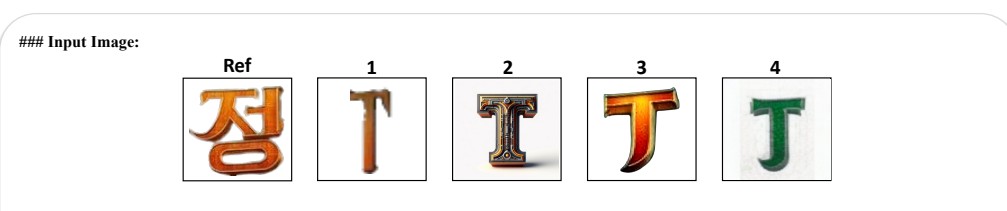

### GPT-4V Judgement:

Score of image 1: 2
•Justification of your rating: The color in image 1 is a gradient of orange and brown, somewhat closer to the wooden tones in reference image but not an exact match. The texture is minimal and does not capture the detailed wooden texture observed in reference image. Moreover, the font style is significantly thinner, lacking the boldness and intricate designs present in the font of reference image.

Score of image 2: 2
•Justification of your rating: Image 2 features metallic orange and gray tones, differing greatly from the wooden color of reference image. Its texture is metallic and industrial, contrasting sharply with the natural wooden texture of reference image. Additionally, the font style in image 2 is more futuristic and mechanical, lacking the traditional wooden style's thickness and sharpness seen in reference image.

Score of image 3: 3
•Justification of your rating: The colors in image 3 consist of a gradient of orange and yellow, slightly resembling the wooden hues of reference image but not perfectly matching. The texture is smoother and less detailed than the wooden texture of reference image. The font style shares a similar thickness and sharpness to that of reference image, but it is simpler and lacks the intricate details.

Score of image 4: 1
•Justification of your rating: Image 4 has a green color, completely different from the wooden color of reference image. Its texture appears pixelated and far less detailed compared to the wooden texture in reference image. Furthermore, the font style is significantly different, displaying a different thickness and a design that diverges from the font style of reference image.

### Claude-3-Opus Judgement:

Score of image 1: 4, The color of the typography in image 1 closely matches the orange-brown hue of the reference image. The texture appears slightly smoother and more uniform compared to the reference, but still maintains a good degree of similarity. The font style, thickness, and sharpness are well-preserved, making the overall typography quite similar to the reference.

Score of image 2: 3, Image 2's typography has a more colorful and vibrant appearance compared to the reference, incorporating blue and yellow tones. The texture seems to have more of a metallic or glossy quality that differs from the reference. However, the font style and general design are still fairly well-matched.

Score of image 3: 4, The typography in image 3 exhibits colors that are quite similar to the reference, with a predominantly orange hue. The texture has a slightly more weathered or distressed look compared to the reference, but this difference is relatively minor. The font maintains good fidelity to the reference in terms of style, thickness, and sharpness.

Score of image 4: 3, Image 4's typography incorporates a green color scheme that deviates from the orange tones of the reference image. The texture appears somewhat smoother and more uniform than the reference. Despite these differences in color and texture, the font style and overall design remain largely consistent with the reference typography.

Figure 10: Judgment about human perspective template from GPT-4V and Claude-3-Opus.

| ### Input Image 1 | ### Input Image 2 | ### Input Image 3 | ### Input Image 4 | ### Input Image 5 |



### GPT-4V Judgement:
[Score between image 5 and target image 1]: 2
Justification:
•Color Consistency: Image 1 features a plain brown tone, differing significantly from the gradient orange and yellow hues in image 5.
•Texture Quality: Image 1 has a flat and smooth surface, lacking the textured, almost metallic sheen seen in image 5.
•Font Fidelity: While the shape of the typography in image 1 is somewhat similar, the lack of texture and depth in color makes it feel quite different from the intricate details of image 5.

[Score between image 5 and target image 2]: 4
Justification:
•Color Consistency: The colors in image 2, which include blue, black, and gold, incorporate a gradient effect similar to the reference, but the specific hues differ.
•Texture Quality: The texture in image 2 is rich and detailed, closely matching the intricate, layered look of image 5.
•Font Fidelity: Image 2 shows high similarity in font style, ornate details, and thickness, very well preserved compared to image 5.

[Score between image 5 and target image 3]: 3
Justification:
•Color Consistency: The gradient colors in image 3 (red to yellow) are somewhat similar but not identical to the orange and yellow gradient in image 5.
•Texture Quality: The glossy texture in image 3 reflects some sheen of image 5 but lacks the same depth and metallic quality.
•Font Fidelity: The overall shape and contour of the font in image 3 is good, but the different texture and color scheme alter the overall feel compared to image 5.

[Score between image 5 and target image 4]: 1
Justification:
•Color Consistency: Image 4's flat green color is vastly different from the gradient orange and yellow in image 5.
•Texture Quality: The texture in image 4 is very flat and lacks the detailed, textured appearance seen in image 5.
•Font Fidelity: The font style in image 4 is simplified and does not capture the ornate thickness and curves of image 5, resulting in low fidelity across all aspects: color, texture, and font style.

### Claude-3-Opus Judgement:

Here is my evaluation of the similarity between the typography in images 1-4 compared to the original typography in image 5:

Score about between image 5 and target image 1: 2, While image 3 maintains the general shape of a "T", it deviates significantly from the original typography. The wooden texture and flat brown color are completely different from the metallic orange and red of image 5. The font is noticeably thicker and lacks the sharp, mechanical edges.

Score about between image 5 and target image 2: 4, Image 2 does an excellent job replicating the intricate mechanical design and shiny metallic texture of the original "T" typography. The colors lean more heavily into orange and yellow tones compared to image 5, but overall it achieves a high degree of fidelity to the source material in terms of style and detail.

Score about between image 5 and target image 3: 3, The color scheme of the "T" in image 1 captures the orange and red gradient of the original, but the texture appears more rough and distressed compared to the smooth metallic surface in image 5. The font style and thickness are fairly well preserved.

Score about between image 5 and target image 4: 3, Image 4 preserves the color scheme quite well, utilizing a similar orange to green gradient as the original. However, the font is slightly thicker and fuzzier compared to the crisp lines in image 5. The felt-like texture is also a departure from the smooth metallic surface of the original.

Figure 11: Judgment about encoding each image template from GPT-4V and Claude-3-Opus.

## A.2 Safety Misclassification of GPT-4V

In fig. 12, we present examples that, despite containing content deemed safe, elicited a "400 Bad Request" error from GPT-4V.

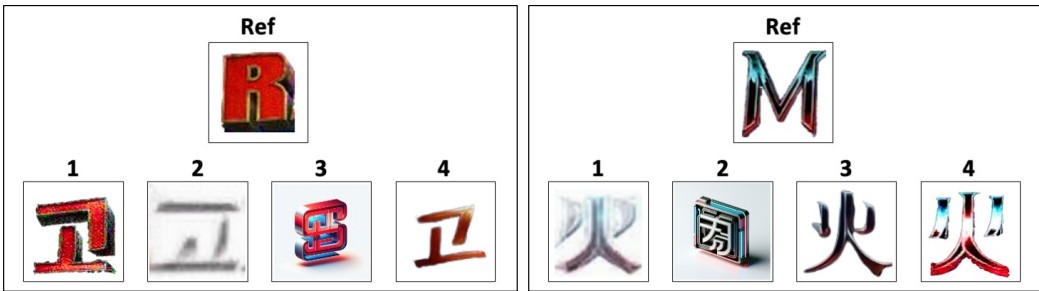

Figure 12: Samples of GPT-4V safety misclassification.

## B  More Ablation Studies

**Visual Text Generation Prompts.**   The prompt for visual text generation can be expressed in various ways, as shown in fig. 13. In this ablation study, the seed was fixed at 100, and the prompt was changed for experimentation. While the initial images generated for each prompt vary greatly, the final generated images at the end of training maintain a variety of style fidelity without losing legibility.

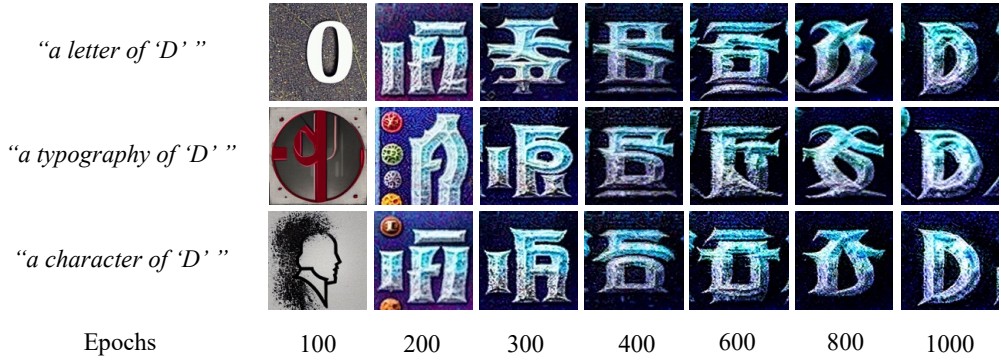

Figure 13: Various prompts for generating visual text 'D' and the corresponding generation results.

## C  Dataset Curation Details

**Collecting Multilingual Film Poster Images.**   We collected multilingual posters for movie titles from a movie database website. Then, we kept only those titles that had pairs in different languages, creating pairs of multilingual posters.

**Filtering by Style Similarity.**   We hired three AI researchers to perform filtering tasks by determining the style similarity of the text in two poster images displayed on the screen. A screenshot of the full text of instructions for this task can be found in fig. 14, which also shows the annotation tool we developed specifically for this task.

**Character-level Bounding Box Annotation.**   The three human annotators who participated in the prior filtering task volunteered for this task as well. They also carried out bounding box labeling on

the filtered images. The instructions for this task are introduced in fig. 14. Each annotator receives a wage ranging from 12-15 dollars per hour, the total amount spent on participant compensation is 648 dollars.

**Instruction:**
- **Selection Task:** Select a similar multilingual visual text image pair
- **Bounding Box Annotation Task:** Apply bounding box labeling to each individual character and the entire text for the selected similar image pair

**Process:**
- **Selection Task**
    1. View the two visual text images displayed on the screen, and if the style of the visual text matches, press 'K' (Keep)
    2. In all other cases, press 'D' (Discard)
    **Note:** Please choose based on the similarity of the visual text, not the overall similarity of the images presented.

- **Bounding Box Annotation Task**
    1. Mark the area of the visual text's each individual character by placing a dot at the top left and bottom right.
    2. Once the areas for each individual character are marked, finally indicate the area for the entire word.

**Example for Similar Image Pair:**

# Example 1

# Example 2

**Example for Bounding Box Annotation:**

# Top left

# Bottom right

# Annotation Result

Individual character

Entire word

Figure 14: A screenshot of the human annotation interface for MuST-Bench dataset curation.

## C.1 Ethical Considerations

In this paper, we present the MuST-Bench, which incorporates copyrighted film posters designed by human experts. To address the issue of copyright, instead of distributing raw data, we provide download links for each poster image along with bounding box annotations. Furthermore, accompanying code is made available, enabling easy transformation of the posters into the format proposed in the paper. This approach ensures compliance with copyright laws while maintaining the utility of the dataset for research purposes.

## D    Implementation and Training Details

SIGIL comprises two main components, the generator and the corrector. The implementation and training details for each are as follows:

**Generator.**   For the pre-training configuration, we employed the runwayml/stable-diffusion-v1-5 model publicly available weights from the Huggingface Hub (https://huggingface.co/models). The training dataset utilized was derived from the MuST-Bench style subject, where each style subject allowed for fine-tuning on two glyph combinations. Training was conducted for approximately 1,000 epochs. The total epochs were adjusted based on the dataset volume; for instance, a dataset containing 15 training samples warranted an extension to 1,005 epochs to ensure thorough model training. The learning rate was maintained at 1e-4 throughout the training process. Regarding the framework and computational resources, we extended the LoRA-based implementation of Dreambooth to process multi-subject inputs, allowing for concurrent fine-tuning on two glyph subjects. Training was performed on a single NVIDIA A100 40GB GPU, with each session completing in about 20 minutes.

**Corrector.**   The corrector was initiated from the last checkpoint obtained after the preliminary fine-tuning phase (generator's last checkpoint). The training dataset comprised images sampled during the generator's operation, serving as the primary data for further training. Training was conducted with an emphasis on efficiency, incorporating an early stopping mechanism to curtail the training as soon as the model reached a satisfactory level of performance. On average, the training duration was approximately 30 minutes using the same GPU as the generator. The learning rate was set to 3e-4, which was determined to be optimal for achieving convergence while maintaining training stability. Additionally, the EasyOCR tool was employed to facilitate multi-language text recognition. While utilizing this off-the-shelf OCR model, we adapted its output mechanism to provide confidence scores for characters presented in the prompt instead of predicted characters.

## E    Image Input to DALL-E3

SIGIL can accept style images as input. However, DALL-E 3 only accepts text inputs. Therefore, to ensure fairness in evaluation, GPT-4V is used as a bridge to enable DALL-E 3 to "see" images. As shown in  fig. 15, GPT-4V views the style image and creates a description of that style. This description is then combined with the textual prompt and sent to DALL-E 3. The importance of using GPT-4V for style description can be seen by comparing fig. 16 (a) and (b).

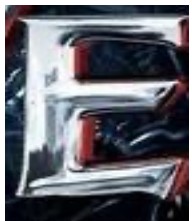

"The typography style in the image is metallic, sleek, and has a futuristic appeal with sharp angles and a glossy finish. The use of a contrasting red accent enhances its modern and dynamic look."

(a) Style image          (b) GPT-4V descriptions for style image

Figure 15: GPT-4V description about style image.

## F    User Studies

In the user study, assessments were conducted on two distinct parameters, style fidelity and legibility. 60 participants were provided with detailed guidelines for each instruction, as outlined in fig. 17, before initiating the study.  fig. 18 is a sample of a user study.  fig. 19 exemplifies a scoring sheet used for ranking the outcomes of comparison methods based on style fidelity, and illustrates a scoring sheet designed for the evaluation of legibility.

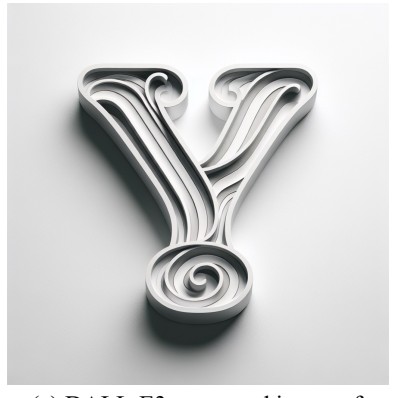
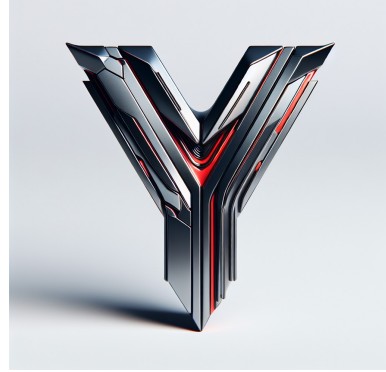

(a) DALL-E3 generated images for "a typography of 'Y'".

(b) DALL-E3 generated images for GPT-4V's style image descriptions + "a typography of 'Y'".

Figure 16: Comparison of DALL-E 3 generated outputs based on the presence of GPT-4V descriptions.

## Multi-language Visual Text Generation User Study

[English]

In this task, two types of questions will alternate.
**# Question 1** (Style Similarity): Look at the reference image and rank images A, B, C, D in order of similarity to the reference image's style.
- Note: Rank them based on style similarity, not personal preference.
- Style includes color, texture, font, etc.

**# Question 2** (Legibility of Text): Based on the given prompt, view the text in images A, B, C, D and rank them in order of accuracy and legibility.

Figure 17: User study instruction and user interface.

## G    Limitations

In the real world, there is abundant high-resolution typography data. However, for collecting multi-language pairs with the same style, movie poster data has proven to be the most effective. When extracting typography from movie posters, the resulting size is relatively small. If the input style image is low resolution, it can complicate VAE-based pixel-level encoding, potentially affecting the generation results. In future research, we aim to explore methods to generate typography in different languages from single-language style images, thereby utilizing more real-world data.

Unlike English, some glyphs with complex strokes in Chinese and Korean present challenges in generation. Applying fine-grained image generation methods could enable the accurate creation of all glyphs.

Lastly, we observed that the EasyOCR model used as a reward model in the Reinforcement Learning process sometimes exhibits False Negative issues with generated images. Additionally, if the model

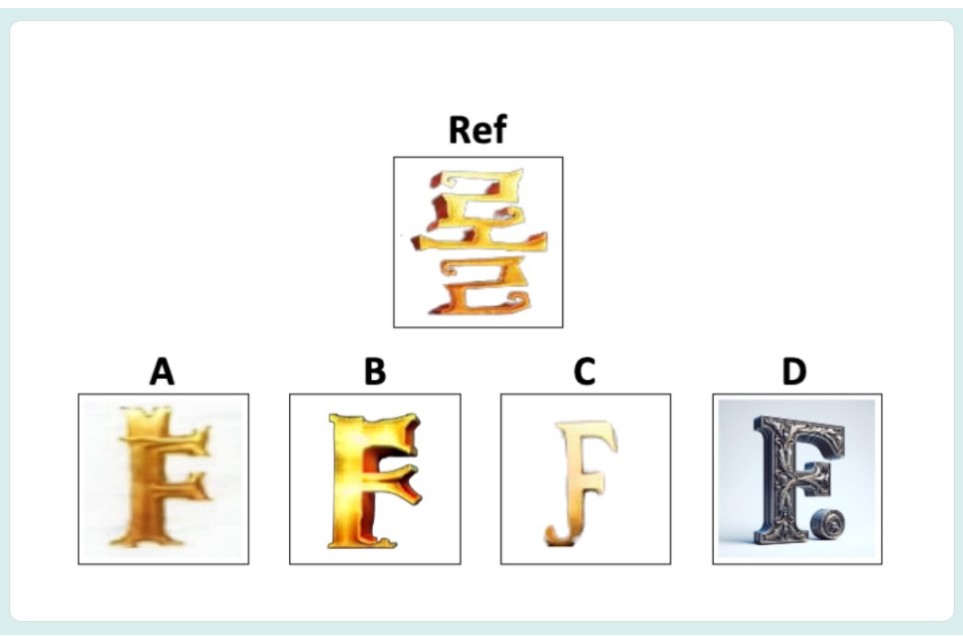

Figure 18: Evaluation sample of the user study.

generates words not in EasyOCR's vocabulary, it can complicate the process. Future work will attempt to use more robust OCR models with larger vocabularies and higher prediction accuracy.

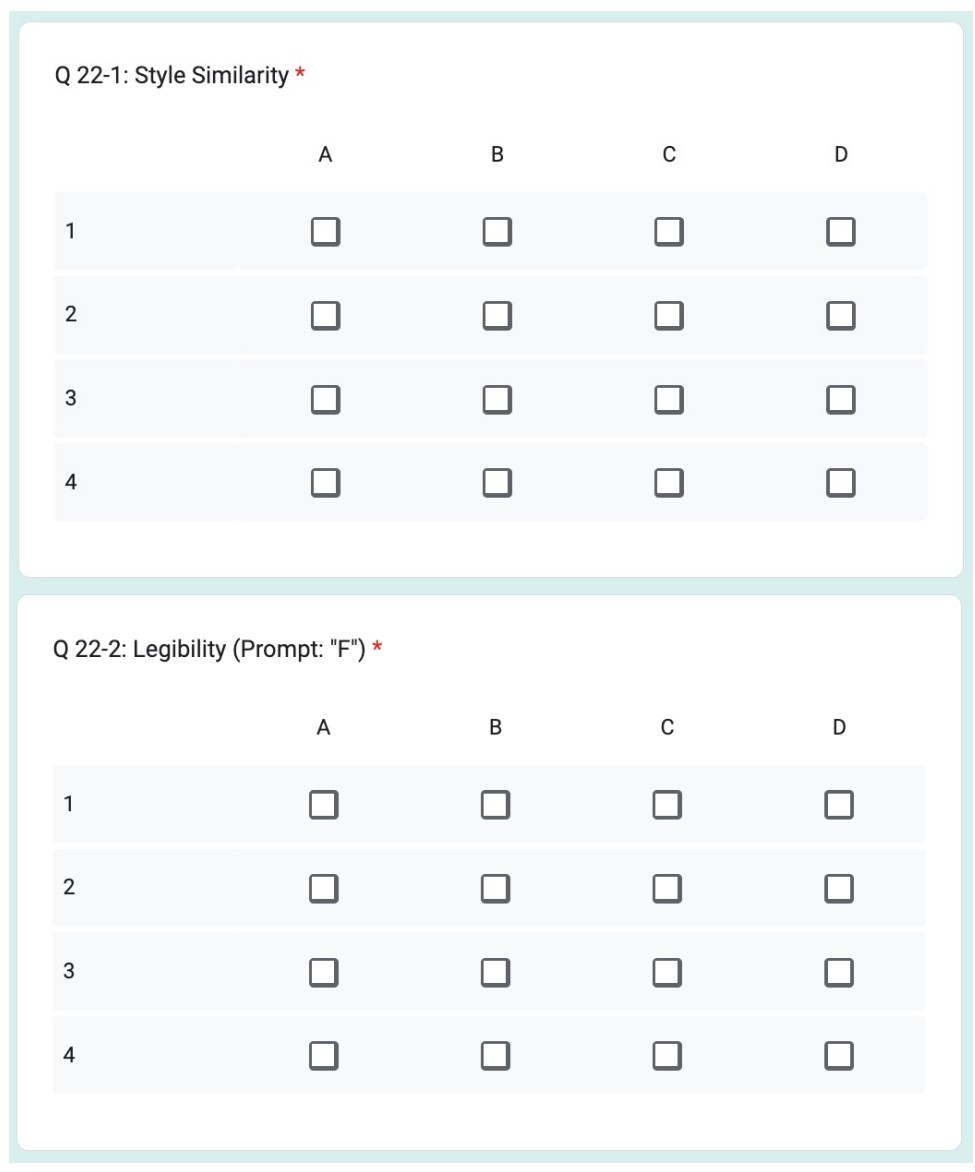

Figure 19: Questionnaire examples of style fidelity and legibility.

