# OpenReview forum: "Towards Visual Text Design Transfer Across Languages"
_NeurIPS.cc/2024/Datasets_and_Benchmarks_Track — NeurIPS 2024 Track Datasets and Benchmarks Poster_

### Official Review · Reviewer_Rme8 · 2024-07-22
**Review by Reviewer Rme8**

**Rating:** 6
**Confidence:** 3

**Review:**

The paper's major contributions are the MuST-Bench dataset and the SIGIL framework for generating stylized glyphs from multiple languages. The MuST-Bench dataset can be quite useful for future researchers, as it provides ground truth glyphs with the same style across many different languages. The proposed SIGIL framework have strong performance in the experiments, both quantitatively and qualitatively, against strong baselines. However, many important information are missing about the methodology of SIGIL, which makes it difficult to accurately assess the merit of SIGIL. The experiments seem to be well-designed, with a comprehensive suite of evaluation metrics.

Pros (see Strengths section for details):

1. The new dataset can be quite useful.

2. The proposed method, SIGIL, has good performance.

3. The experiment contains a comprehensive set of metrics.

Cons (see Opportunities for Improvement for details):

1. The methodology description of SIGIL is very unclear, with many important information missing.

2. There are no quantitative results on ablation studies and no ablations on the inclusion of objectives like $L_{glyph}$ and PPO.

**Strengths:**

1. The new dataset can be quite useful. It contains glyphs from different languages with almost the same styles, allowing supervised training over style-guided glyph generation, and can serve as ground-truth during evaluation.

2. The proposed method, SIGIL, has good performance. The generated images looks really good, and it out-performed baselines quantitatively in experiments.

3. The experiment contains a comprehensive set of metrics, including CLIP score, OCR, MLLM eval and user study.

**Additional Feedback:**

N/A

**Clarity:**

All parts except the SIGIL methodology is well-written. The SIGIL methodology has many missing details, wrong formulas, and confusing descriptions (see "Opportunity for Improvement" section).

**Correctness:**

It is difficult to assess the correctness of SIGIL, since many crucial details about its training objective and inference process are missing. The evaluation method and experiment design seem appropriate.

**Documentation:**

Documentation is sufficient.

**Ethics:**

No ethical concern.

**Limitations:**

Discussion is adequate (section G in Appendix)

**Opportunities For Improvement:**

1. One major problem with this paper (in its current state) is that the methodology description of SIGIL is very unclear:

(1) The paper did not clearly indicate what exactly was the inputs to SIGIL during training and inference. For example, let's assume we want to generate a glyph for letter "K" in a specific style, and we have style images "A", "B", "C", "D" in that style as well as ground truth image (i.e. "K" in the desired style). Then during training, is ground truth "K" or "ABCD" used as input to the forward process of the diffusion? Do you train your model on just one style at a time, then perform inference? Or do you train your model jointly with multiple styles, then provide a specific style (with style images ABCD) for inference? Do you use the ground truth "K" at all during training?

(2) The training losses are confusing. If $\epsilon$ indicates noise, then the glyph loss in Eq (2) makes no sense since it is the difference between a predicted noise and a latent representation $z$. Also, for $L_{diff}$, in Fig 4 it seems like it is between multiple style images and one generated image (given text prompt to generate a different glyph), how does that work (Eq (1) only works for one input)?

2. There are no quantitative results on ablation studies. Also, the existing ablation studies was only on the hyperparameters and text prompt design, and the paper could benefit from ablations on objectives (i.e. SIGIL without $L_{glyph}$, or SIGIL without the RL part) demonstrating the necessity of each component.

**Relation To Prior Work:**

Discussion is adequate.

**Summary And Contributions:**

In this paper, the authors first presented a dataset called MuST-Bench that contains same-style glyphs across multiple languages. These glyphs are mined and annotated from the same film posters translated across languages. In addition, the authors trained a model called SIGIL using data from MuST-Bench that can generate glyphs from different languages from text prompts, given examples of glyphs within that style.

---

> ### Author Rebuttal · Authors · 2024-08-17
>
> We appreciate the reviewer's acknowledgment of the usefulness of our dataset and effectiveness of our method. Below, we address the points raised by the reviewer:
>
> **Point 1-1. Clarifying method description**
>
> We acknowledge that our method should be described clearer and promise to update the final draft to reflect the following descriptions:
> During the training process, we use 3 to 5 specific style images (a common number used in personalized image generation [1, 2]), along with a prompt (e.g., "a typography of 'K'") and a glyph image (e.g., a white background with a black 'K' glyph created using the Pillow library).
> However, the ground truth (e.g., the desired style image of 'K') is not used for training. Since we assume there is no ground truth in real-world translation tasks, we introduce a SIGIL method instead of relying on direct supervision from the ground truth, which enables us to generate the glyph.
> During inference, only the prompt (e.g., "a typography of 'K'") is used as input.
> We train our model on one style with multiple glyphs. As shown in Appendix G, while previous approaches could only generate one glyph per style, we have extended this capability to two glyphs. For example, we can generate both 'A' and 'B' in the 'vivid style.' The training time for this can be found in Appendix D.
>
> [1] DreamBooth: Fine Tuning Text-to-Image Diffusion Models for Subject-Driven Generation
>
> [2] An Image is Worth One Word: Personalizing Text-to-Image Generation using Textual Inversion
>
> [3] DS-Fusion: Artistic Typography via Discriminated and Stylized Diffusion
>
> **Point 1-2-1. Training loss: $L_{glyph}$**
>
> The glyph loss effectively controls how much of the glyph's shape characteristics are embedded in the generated output latent $x_{0}$, leading to a glyph-shaped alphabet. Below are detailed explanation:
>
> In the original DDPM [1], the reverse (generation) process is described by the following equation in Line 4 of Algorithm 2:
>
> $x_{t-1} = \frac{1}{\sqrt{\alpha_t}} \left( x_{t} - \frac{1 - \alpha_t}{\sqrt{1 - \overline{\alpha_t}}} \times e_{\theta}(x_t, t) + \sigma_t \times z \right)$
>
> In this equation, $e_{\theta}(x_t, t)$ is the model's predicted noise, $\sigma_t$ is a time-dependent noise coefficient, and $z \sim \mathcal{N}(0, I)$ is a Gaussian noise sampled at each timestep.
>
> Our loss in Equation 3 is as follows:
>
> $\text{Loss} = |e_{\theta} - e|^2 + f(t) \times |e_{\theta} - z_{\text{glyph}}|^2$
>
> By *Triangle Inequality*, it becomes:
>
> $\|e_{\theta} - e + f(t) \times (e_{\theta} - z_{\text{glyph}})\|^2 \leq \|e_{\theta} - e\|^2 + \|f(t) \times (e_{\theta} - z_{\text{glyph}})\|^2$
>
> So, our U-Net predicts $e_{\theta} = \frac{1}{1 + f(t)} e + \frac{f(t)}{1 + f(t)} z_{\text{glyph}}$ and $f(t) << 1$ so, $e_{\theta} \sim e + f(t) z_{\text{glyph}}$.
>
> With this, we can rewrite Line 4 of Algorithm 2 in DDPM as:
>
> $x_{t-1} = \frac{1}{\sqrt{\alpha_t}} \left( x_{t} - \frac{1 - \alpha_t}{\sqrt{1 - \overline{\alpha_t}}} \times (e + f(t)  z_{\text{glyph}}) + \sigma_t \times z \right)$
>
> Thus, we can express $x_{t-1}$ as:
> $ x_{t-1} = \text{sampling}(e_{\theta}, DDPM^{\text{wo-glyph}}) + \text{glyph-contribution}(f(t) \times z_{\text{glyph}}, DDPM^{\text{w-glyph}}) $
>
> With the above-derived equation, $x_{t-1}$ is gradually adjusting the glyph latent to influence the final output latent, allowing the model to learn and modify the shape characteristics of the glyph ( trace of glyph, not a whole glyph) during the generation process.
>
> [1] Denoising Diffusion Probabilistic Models
>
> **Point 1-2-2. Training loss: $L_{diff}$**
>
> We apologize that Fig 4 does not correctly convey our concept.
> In the SIGIL process, the steps are as follows: (1) The generator is trained independently first, and (2) the corrector is then trained by resuming from the generator's checkpoint. Although the corrector utilizes the generator's checkpoint during its training, the generator's training remains entirely unaffected by the corrector.
> We have redrawn Figure 4 to clearly distinguish the actions of the generator and corrector, as well as to clarify the data flow. The updated Figure 4 is included in the attached PDF.
>
> **Point 2. More ablation studies**
>
> We share the reviewer’s concern about the lack of ablation study and augmenting our experiments now. All numerical results are in the separate comment above.
>
> First, we report quantitative results, which re-affirms our findings in the qualitative studies. Second, we conduct more ablation on the objective part. Specifically, we experiment applying SIGIL without $L_{glyph}$, SIGIL without the Generator(combine style & glyph), and SIGIL without the Corrector(RL part) to examine the contribution of each component.
>
> The results in the table above show that the generator has the greatest impact on style generation accuracy, while glyph generation accuracy is most significantly influenced by
> $L_{glyph}$​. Additionally, the Corrector primarily proves effective when the generator already achieves a certain level of success in glyph generation.

---

> ### Author Rebuttal · Authors · 2024-08-17
>
> Quantitative results for ablation studies:
>
> (1) Hyperparameters
>
> | 𝜆           | OCR    | CLIP-I |
> |-------------------|--------|--------|
> | 0.5     | **0.9423** | 0.8565 |
> | 0.1     | 0.7163 | 0.8673 |
> | 0.01  | 0.0038 | **0.8855** |
>
> |  𝜏          |   OCR  |  CLIP-I  |
> |-----------------|--------|----------|
> | 100  | 0.0663 | **0.8696**   |
> |  50   | 0.7163 | 0.8673   |
> | 10   | **0.9276** | 0.8603   |
>
> (2) Module ablation (OCR & CLIP Eval)
>
> | Method                        | OCR (English) | OCR (Chinese) | OCR (Korean) | CLIP-I (English) | CLIP-I (Chinese) | CLIP-I (Korean) |
> |-------------------------------|---------------|---------------|--------------|------------------|------------------|-----------------|
> | SIGIL (w/o glyph loss)         | 0.001         | 0.001         | 0.002        | **0.8864**       | **0.8833**       | **0.8834**      |
> | SIGIL (w/o Corrector)          | 0.4529        | 0.3962        | 0.2188       | 0.8674           | 0.8617           | 0.8694          |
> | SIGIL (w/o Generator)          | 0.18          | 0             | 0            | 0.8078           | 0.801            | 0.804           |
> | **SIGIL**                      | **0.7163**    | **0.7481**    | **0.6577**   | 0.8673           | 0.8618           | 0.8692          |
>
> (3) Module ablation (MLLM Eval)
>
> | Evaluator                  |             |           GPT-4V           |                      |             |           Claude           |                      |
> |----------------------------|------------------|----------------------|----------------------|--------------------|----------------------|----------------------|
> |                            | English          | Chinese              | Korean               | English            | Chinese              | Korean               |
> | SIGIL (w/o glyph loss)      | 2.5              | 2.6                  | 2.5                  | 2.9                | 3.2                  | 2.6                  |
> | SIGIL (w/o Corrector)       | 2.5              | 2.6                  | 3.1                  | 3.3                | 3.1                  | 2.6                  |
> | SIGIL (w/o Generator)       | 2                | 1.3                  | 1.2                  | 2.1                | 2.4                  | 2.3                  |
> | **SIGIL**                   | **4.1**          | **3.3**              | **3.8**              | **3.4**            | **3.2**              | **3.6**              |

---

> > ### Comment · Reviewer_Rme8 · 2024-08-28
> > **Score increased**
> >
> > Thank you very much for your clarifications! You have addressed my concerns and I have increased my score to positive.

---

> ### Comment · Area_Chair_Hr7U · 2024-08-27
> **[urgent] AC requesting reviewer to comment**
>
> Dear Reviewer,
>
> The author-reviewer discussion period is ending imminently. Please respond to the authors’ rebuttal ASAP and confirm that you have read their response.
>
> Thank you,
> AC

---

### Official Review · Reviewer_5qTD · 2024-07-25
**Review For Towards Visual Text Design Transfer Across Languages**

**Rating:** 7
**Confidence:** 3
**Correctness:** The dataset is constructed in a sound…
**Clarity:** The paper is well written.

**Review:**

Pros:

- Understudied problem and the authors thoroughly address this by introducing a benchmark, dataset + method. The paper discusses the benchmark in details.
- The method for target sequence generation (With consistent style) is simple and well thought. However, I have some comments on it (see Weaknesses).

Cons:

- While the benchmark is a good starting point, it might be too small to be interesting to the community. The authors are suggested to scale up the size of the dataset.
- For evaluation, the authors are suggested to motivate more why CLIP is a good choice for evaluating image to image similarity compared to other models. Also can two function to CLIP and the MLLM be circumvented through one call?
- One can think of this problem as image-conditioned text-based editing or image-conditioned mask + text based editing. The paper is missing this baseline.


Overall the authors study an important problem and take a good step towards solving it. It would be great if the authors can get back regarding the Cons during the rebuttal to maintain my score (I am giving a slightly high score, given the practicality of the problem).

**Strengths:**

Check Pros / Strengths above.

**Additional Feedback:**

Check cons!

**Documentation:**

There's a slight lack of documentation. The authors are suggested to provide an anonymized link for the dataset.

**Limitations:**

The dataset presented is small! There might be limited interest to the community corresponding to it.

**Opportunities For Improvement:**

I would suggest the authors to add more baselines to compare their method against.

**Relation To Prior Work:**

The paper has discussions to prior work.

**Summary And Contributions:**

The submission studies the challenging task of generating typography with a consistent style across various languages. The paper thoroughly introduces a benchmark, dataset and a method to achieve this goal. First the MustBench dataset can be used for evaluating multimodal style translation for typography images and also introduces a novel training framework to achieve this.  Overall, the problem is very practical and timely, and the authors provide a comprehensive framework for it.

---

> ### Author Rebuttal · Authors · 2024-08-17
>
> We appreciate the reviewer's acknowledgment of the importance of our task, noting it as an understudied area. We also value the feedback provided on our method. Below, we address the points raised by the reviewer:
>
> **Point 1. Dataset size**
>
> | While the benchmark is a good starting point, it might be too small to be interesting to the community. The authors are suggested to scale up the size of the dataset.
>
> We acknowledge the reviewer's concern and have significantly expanded our MuST-bench dataset by more than double during the rebuttal period, increasing the sample count from 460 to 1,127. The extended dataset can be accessed at the anonymized Huggingface repository (https://huggingface.co/datasets/ananymousaccount/MuST-Bench). With this expansion, our dataset is now comparable in size, or even larger, than other widely used multimodal benchmarks, such as Winoground (400 samples [1]) and Drawbench (200 samples [2]). The updated experimental results will be included in the final draft.
>
> [1] Thrush, Tristan, et al. "Winoground: Probing vision and language models for visio-linguistic compositionality." Proceedings of the IEEE/CVF Conference on Computer Vision and Pattern Recognition. 2022.
>
> [2] Saharia, Chitwan, et al. "Photorealistic text-to-image diffusion models with deep language understanding." Advances in neural information processing systems 35 (2022)
>
> **Point 2. CLIPScore evaluation**
>
> | For evaluation, the authors are suggested to motivate more why CLIP is a good choice for evaluating image to image similarity compared to other models. Also can two function to CLIP and the MLLM be circumvented through one call?
>
> Thank you for the insightful feedback. While using CLIP to measure image-to-image similarity does not follow straight from its training objective, it empirically shows good correlation with human decisions and hence are widely used in domain similar to ours [1,2,3]) for automatic evaluation. We promise to augment this result with another image-to-image metric based on single modal model (DINO [4]) in the final draft.
>
> Theoretically, MLLMs can subsume CLIP as a multimodal evaluator. However, there are two implementational advantages of CLIP: 1. CLIP is a much ligher model, and hence can offer cost-effective evaluation metrics for measuring simple image-to-image similarity. 2. MLLMs are not very reliable at assigning scores, so the two models work complementarilly to each other. (M)LLMs are known to be affected by a number of irrelevant biases (e.g. order of options [5]), so they should at least in the current status be complemented by simpler and more interpretable models, i.e. CLIP in this case.
>
> [1] DreamBooth: Fine Tuning Text-to-Image Diffusion Models for Subject-Driven Generation
>
> [2] GlyphControl: Glyph Conditional Control for Visual Text Generation
>
> [3] DS-Fusion: Artistic Typography via Discriminated and Stylized Diffusion
>
> [4] Oquab, Maxime, et al. "Dinov2: Learning robust visual features without supervision." arXiv preprint arXiv:2304.07193 (2023).
>
> [5] https://www.lesswrong.com/posts/S4aGGF2cWi5dHtJab/your-llm-judge-may-be-biased
>
> **Point 3. Additional baseline**
>
> | One can think of this problem as image-conditioned text-based editing or image-conditioned mask + text based editing. The paper is missing this baseline.
>
> Thank you for the great suggestion! We now test the suggested baseline against our benchmark. Specifically, we utilized the Stable Diffusion 1.5 image editing pipeline, consistent with the pretrained diffusion model employed in our SIGIL method.
>
> As shown below, the baseline models exhibited poor performance, primarily because they are not fine-tuned for character generation tasks. A qualitative examination of the generated outputs further supports this, as the baseline frequently fails to produce recognizable characters.
> While the SD 1.5 inpainting approach achieved slightly higher scores in CLIP-I and MLLM evaluations, this comes at a high expense of the OCR score. We manually examined the outputs to confirm that SD 1.5 inpainting usually generates non-recognizable glyphs. We have included a qualitative results figure in the supplementary material to illustrate these points.
>
> | Method      | OCR (English) | OCR (Chinese) | OCR (Korean) | CLIP-I (English) | CLIP-I (Chinese) | CLIP-I (Korean) |
> |-------------|:-------------:|:-------------:|:------------:|:----------------:|:----------------:|:---------------:|
> | SD (t2i)    | 0.0154        | 0             | 0            | 0.8169           | 0.8136           | 0.8137          |
> | SD (i2i)    | 0.0067        | 0             | 0            | 0.8677           | 0.8588           | 0.8641          |
> | SD (mask)   | 0.0365        | 0             | 0            | **0.8746**       | **0.875**        | **0.8748**      |
> | SIGIL       | **0.7163**    | **0.7481**    | **0.6577**   | 0.8673           | 0.8618           | 0.8692          |
>
> | Evaluator | | GPT-4V | | | Claude | |
> |---|---|---|---|---|---|---|
> | | English | Chinese | Korean | English | Chinese | Korean |
> | SD (t2i) | 1.22 | 1.38 | 1.29 | 2 | 2.9 | 2.3 |
> | SD (i2i) | 1.33 | 1.25 | 1.57 | 1.7 | 1.6 | 2.1 |
> | SD (mask) | 2.89 | 3 | 3.43 | 3.5 | 3 | 3.1 |
> | SIGIL | **3.67** | **3.5** | **3.43** | **3.7** | **3.4** | **3.3** |
>
> **Point 4. Better data documentation**
>
> | There's a slight lack of documentation. The authors are suggested to provide an anonymized link for the dataset.
>
> We apologize for the lack of documentation in the attached files. For better ease of access, we provide the anonymized link in Hugginface Dataset format here: https://huggingface.co/datasets/ananymousaccount/MuST-Bench, which includes a user-friendly data viewer. This allows for easier exploration and understanding of the dataset directly within the platform. The dataset provided here is identical to the one previously uploaded on the OpenReview interface and includes the scale-up of the data as mentioned in Point 1 of our rebuttal.

---

> ### Comment · Area_Chair_Hr7U · 2024-08-27
> **[urgent] AC requesting reviewer to comment**
>
> Dear Reviewer,
>
> The author-reviewer discussion period is ending imminently. Please respond to the authors’ rebuttal ASAP and confirm that you have read their response.
>
> Thank you,
> AC

---

### Official Review · Reviewer_5Mds · 2024-07-26

**Rating:** 7
**Confidence:** 3
**Correctness:** Yes
**Clarity:** Yes

**Review:**

This paper presents MuST-Bench and SIGIL, making good contribution to the task of visual text translations in multi-lingual setting. It is an under-represented benchmark category and has originality.

**Strengths:**

- The problem is important.
- The paper is well written, with good and clear figures.
- The results are good compared to previous SOTA.

**Additional Feedback:**

Please see above review.

**Documentation:**

Yes

**Limitations:**

The reviewer does not think there is major limitations beyond the above one.

**Opportunities For Improvement:**

As brought up in the limitations section, it will be interesting to see how a more robust OCR model can help.

**Relation To Prior Work:**

Yes

**Summary And Contributions:**

The paper presents MuST-Bench  to evaluate visual text translations across different languages, where current models struggle with. The authors then propose SIGIL, a new framework that improves image generation using glyph latents, pre-trained VAEs, and an OCR model with reinforcement learning. It outperforms existing methods in style consistency, legibility and visual similarity.

---

> ### Author Rebuttal · Authors · 2024-08-17
>
> We thank the reviewer for recognizing importance of our task and effectiveness of our method. Below are our responses to the points raised by the reviewer:
>
> **Point 1. More robust OCR model**
>
> | As brought up in the limitations section, it will be interesting to see how a more robust OCR model can help.
>
> Thank you for your valuable feedback and great suggestion! We agree that incorporating more robust OCR models could further enhance the performance of SIGIL. To address this, we commit to augmenting our experimental results with more robust OCR modules such as MMOCR [1] and PP-OCRv3 [2] until publication.
>
> There is an important consideration when replacing the OCR module in our setup. We intentionally use different OCR models for training and inference (EasyOCR) versus evaluation (PP-OCR) to ensure a fair comparison with other methods. Consequently, any experiments involving alternative OCR models would require re-evaluating all baselines with the new OCR evaluator to maintain fairness across tested models. Given that these time-intensive experiments cannot be completed within the rebuttal period, we have to postpone presenting the actual results until the final draft.
>
> [1] MMOCR: A Comprehensive Toolbox for Text Detection, Recognition and Understanding
>
> [2] PP-OCRv3: More Attempts for the Improvement of Ultra Lightweight OCR System

---

> ### Comment · Area_Chair_Hr7U · 2024-08-27
> **[urgent] AC requesting reviewer to comment**
>
> Dear Reviewer,
>
> The author-reviewer discussion period is ending imminently. Please respond to the authors’ rebuttal ASAP and confirm that you have read their response.
>
> Thank you,
> AC

---

### Decision · Program_Chairs · 2024-09-26

**Decision:**

Accept (Poster)

**Comment:**

This paper introduces MuST-Bench, a benchmark for Multimodal Style Translation: generating typography with a consistent style across various languages. Data is sourced from film-poster pairs using English as the pivot language and Chinese, Korean, Thai, Russian, and Arabic as target languages. Because these languages have a distinct writing system from English, typography translation is non-trivial. To help models perform well on MuST-Bench, this paper also introduces the Style Integrity and Glyph Incentive Learning (SIGIL) framework for learning that augments existing diffusion models with feedback signals to better succeed at this task.

This work was positively received by the reviewers and after the rebuttal obtained high scores. The reviewers agree the contribution of MuST-Bench is timely, interesting, and novel. Sourcing data from film-poster pairs brings a unique dataset to the community that includes instance-level annotations. The evaluation of existing approaches on this dataset is comprehensive and reviewers agree that the metrics used are sensible. Though SIGIL is clearly engineered to do well at this task, it is relatively agnostic to the precise diffusion model used, and makes an interesting contribution in facilitating progress on MuST-Bench.

Most of the reviewer's concerns were addressed during the rebuttal, including the addition of more data, a new baseline, and various explanations of missing details where applicable. Concerning the latter I urge the authors to update the next revision of their paper to include these. The only outstanding concern that I see is that this dataset focuses on a particular niche, and it is hard to predict whether this task will be picked by the community. That said, I find that the novelty and uniqueness of the data makes up for this considerably, and that this work makes a valuable contribution to the community due to how well it is executed. I recommend accept.